# Evaluation of usability and acceptability of a Peruvian telemental health service for early assessment among vulnerable occupational workers: Mixed-method study with a user-centered design approach

Jimmy Andreyvan Cainamarks-Alejandro [1], Liliana Cruz-Ausejo[1], Miguel Angel Burgos-Flores [1], Jaime Rosales-Rimache[1], Jonh Astete-Cornejo[1], David Villarreal-Zegarra [2]*

1 Centro Nacional de Salud Ocupacional y Protección del Ambiente para la Salud, Instituto Nacional de Salud, Lima, Peru, 2 Universidad Científica del Sur, Lima, Peru

* dvillareal@cientifica.edu.pe; david.villarreal@digitalhealth.pe

## Abstract

### Background

The COVID-19 pandemic marked an increase in depressive, anxiety, and post-traumatic stress disorder symptoms, more specifically among healthcare workers, teachers, and police officers. These workers face external and occupational factors which had a significant impact on mental health, significant increase in workload and direct exposure to the virus, shortage of personnel protective equipment, and instances of abuse, including discrimination. Mental health care in primary care requires a process of early identification and timely referral of complex cases. Tele-health emerges as an effective alternative for addressing challenges in mental health care, although its implementation encounters obstacles.

### Objective

To design a telehealth service that facilitates screening, initial management, and timely referral for mental health diagnoses in workers with prior SARS-CoV-2 infection, and to evaluate usability, acceptability, and user satisfaction.

### Methods

Mixed-method study with a user-centered design approach involving key external and internal service users in three sequential stages (pre-design, co-design, and post-design). The study phases lasted 6 months, involving a total of 23 participants in the pre-design phase (contextual inquiry and preparation and training), 12 participants in the co-design phase (framing the issue, generative design, and sharing design), and

**Data availability statement:** The database is available at https://doi.org/10.6084/m9.figshare.27857520.v1.

**Funding:** We express our gratitude to the Spanish Agency for International Development Cooperation (AECID, in spanish) and the National Program for Scientific Research and Advanced Studies (PROCIENCIA-CONCYTEC-Peru, in spanish), who funded the project, enabling its execution, under contract number 076-2021-PROCIENCIA.

**Competing interests:** The authors have declared that no competing interests exist.

**Abbreviations:** CENSOPAS, National Center for Occupational Health and Environmental Health Protection, National Institute of Health; CSUQ, Computer System Usability Questionnaire Scale; ESCOMA, Outpatient Medical Consultation Users´ Satisfaction Scale; PHQ-9, Patient Health Questionnaire (depressive symptoms); GAD-7, Generalized Anxiety Disorder (Questionnaire for Anxiety symptoms); PCL-5, Posttraumatic Stress Disorder Checklist for DSM-5.

in the post-design phase, 4 participants were involved in service implementation, and 81 participants—drawn from the subgroup of 134 users who received psychoeducation—were included in the efficacy assessment.

## Results

The proposal included the development and evaluation of a service model guide and a telehealth software platform. First, the participants took part in a series of workshops (Pre-design, Co-design) where they provided ideas for meeting the product requirements, based on the Design Thinking methodology framework. The telehealth service model was named TelePsico CENSOPAS. It comprised four processes: a) Service promotion; b) User pre-identification; c) Appointment management; d) Psychoeducation counseling and referral. The Telehealth platform was designed through three cycles of an iterative process and integrated a proprietary development platform with third-party service technologies for communication support and information exchange. During post-design, the pilot test involved 698 screened patients; 193 were identified with mental health risks, and 134 of them received psychoeducation sessions. In addition to user acceptance, the usability score of the platform was $86.1 \pm 16.9$ SD, satisfaction dimensions of the service was $45.1 \pm 7.2$ SD for satisfaction with care processes, and $36.7 \pm 5.2$ SD satisfaction with psychological care.

## Conclusion

The proposal for mental health telehealth services and its supporting platform was successfully developed and accepted by both internal and external users, particularly within well-structured occupational health services in workplaces serving vulnerable occupational groups. In addition, it achieved higher satisfaction and usability scores than Peru's outpatient care services. These findings support the replicability of user-centered design frameworks—such as design thinking—within the occupational health sphere.

## Introduction

The COVID-19 pandemic has increased the incidence of mental health issues, including depression, anxiety and post-traumatic stress disorder symptoms related to mourning [1,2]. During the first year of the pandemic, a global increase of 28% and 26% in symptoms of depression and anxiety, with approximately 159 million additional cases of both conditions, was evident [3]. Furthermore, it has been observed that individuals with a history of SARS-CoV-2 have a 2.5 to 3 times higher risk of experiencing these disorders, compared to the general population, along with a more significant burden of long-term illness [4–7].

Measures to stop the virus from spreading, such as social distancing and remote work, drastically altered working conditions in different occupational sectors [8,9]. Vulnerable occupational groups, such as healthcare workers, police enforcement

personnel, and teachers, experienced a significant increase in workload, direct exposure to the virus, shortage of personnel protective equipment, and instances of abuse, including discrimination [10–16], which had a significant impact on mental health.

In the Peruvian context, there has been a reported prevalence of depression and anxiety of 16.2% and 27.3% among teachers, 19.6% and 17.3% in police officers, and 24.6% and 39.1% in healthcare workers, respectively, during the pandemic. Therefore, the sequels in these vulnerable occupational groups might represent a higher burden of professional illnesses in its economic sector [14,15,17].

The assessment of mental health at work is the primary strategy to mitigate the burden of mental health problems. The World Health Organization (WHO) indicates that workplaces are essential targets for mental health prevention and promotion programs [18]. Likewise, the evidence supports this statement by reporting that workplaces are conducive spaces for implementing strategies and interventions that promote mental health care, as well as prevent, identify, and manage mental health problems effectively [16,19].

Despite the availability of effective treatments, many individuals with mental health problems do not receive adequate care due to internal and external barriers [20]. Personal barriers include stigma and negative attitudes towards mental health, which delay the seeking of help. On the other hand, external barriers, such as lack of access to specialized services, long waiting times, and a shortage of qualified personnel hinder treatment access. Mental health care in primary care requires a process of early identification (screening) and timely referral of complex cases [21]. However, in developing countries like Peru, the normative approach focuses more on specialization than early detection, further overloading the system. Therefore, it is necessary to implement efficient screening and referral processes, thus facilitating access to appropriate treatment [22,23].

In this context, mental health intervention programs, including telehealth services, have been demonstrated to be workable and cost-effective, facilitating widespread access by shortening wait access and ensuring secure and confidential environments for conversation [24,25]. Telehealth has emerged as a promising solution to address the growing mental health care crisis, offering numerous benefits and improving access to care [26,27]. Research indicates that telehealth is effective in treating a wide range of mental health conditions across various patient populations and care settings [28]. Telehealth-based interventions for screening and treatment represent a viable alternative for addressing common mental health issues [24,25,29]. It has been shown to reduce hospital admissions, enhance quality of life, and provide round-the-clock remote access to mental health providers [27]. Telehealth modalities, including video conferencing, mobile apps, and telephone consultations, offer a novel and far-reaching alternative to traditional therapy [28].

Telehealth emerges as an effective alternative for addressing challenges in mental health care, although its implementation encounters obstacles. Successful implementation of telehealth programs requires addressing barriers such as training, technology access, reimbursement, regulations, and program oversight. Despite these challenges, telehealth remains a valuable tool in alleviating the mental health burden, particularly in underserved areas [28]. The variability in the effectiveness of digital health services, depending on geographic location, underscores the influence of sociocultural aspects on their deployment [30,31]. Therefore, using frameworks based on center user design, evaluating user usability and satisfaction are crucial to ensure the success and scalability of these interventions [32–35].

Low- and middle-income countries are particularly susceptible to digital interventions, often validated and designed in high-income settings, not being implemented effectively. Although frameworks exist that propose the implementation of both synchronous and asynchronous digital mental health interventions [36–38], these are predominantly based on studies conducted in high-income countries. Therefore, user-centered evaluations, such as usability and acceptability assessments, are essential to determining the feasibility of future large-scale implementations [39]. This is further complicated by the considerable heterogeneity among healthcare systems in low- and middle-income countries [40]. In the

 

Peruvian context, private telehealth platforms offer integrated services across various aspects of healthcare, including mental health. However, the oversupply of services may compromise coverage and efficiency, especially in the early identification of mental health issues [23,24]. Peruvian regulations define telehealth as providing healthcare services through communication and information technologies, facilitating their organization and use [41]. Although there was a significant increase in demand for telehealth services during the pandemic, its implementation was primarily focused on treating patients, leaving a gap regarding early identification through screening and the initial management of patients to prevent the progression of the illness [42].

The knowledge gaps this study aims to address is that, although telehealth services have been shown to improve access to mental health care, their design often lacks a user-centered approach tailored to the specific needs of high-stress occupational groups. This limitation reduces user acceptance and long-term engagement with these services. Our study addresses these gaps by developing a telehealth service focused on early detection, initial management, and structured referral specifically for these workers in a developing country context such as Peru, where structural challenges magnify the need for efficient mental health interventions.

Therefore, our study addresses the findings of a digital health intervention in the early identification of mental health problems related to symptoms of depression, anxiety, and post-traumatic stress in post-COVID-19 workplaces in Peru, targeted at vulnerable groups such as health workers, police officers, and teachers. Formative research was conducted that described the development and the outcomes of usability and acceptability assessment of a mental health telehealth service based on user-centered design approach. Our Paper describes the results and the methodology applied, providing a local perspective for future occupational interventions that are relevant to Latin America.

## Methods

### Setting

The Peruvian healthcare system is fragmented, with distinct subsystems—such as the Ministry of Health, EsSalud, the military and police health services, and private insurers—that lack interoperability and integration [43]. Furthermore, only 15% of primary care facilities have access to electronic patient records, indicating a low adoption level of key technological tools by the health system and its workforce [44].

Peru's multilingual and multicultural landscape also introduces unique challenges, including cultural stigma surrounding mental health, limited mental health literacy, and linguistic diversity, which includes indigenous languages. These factors create significant barriers to effective digital mental health service delivery. Consequently, user-centered approaches in the design and deployment of telehealth interventions are essential to ensure accessibility and acceptability for Peru's diverse populations.

The TelePsico CENSOPAS system was implemented at a metropolitan scale, specifically in Lima and Callao. The platform targeted users within these urban areas and was designed considering the structural fragmentation of the Peruvian healthcare system and the demographic, cultural, and organizational characteristics of the capital region.

### Design

Formative mixed-study design was organized in three phases (pre-design, co-design, and post-design), based on and adapted from "A generative co-design framework for healthcare innovation: development and application of an end-user engagement framework" by Bird *et al.* [35] and "Health Design Thinking methodology - Creating Products and Services for Better Health" by Ku B *et. al.* [34] (See Fig 1). This type of study allows for the development of a holistic and user-centered approach from conception to the evaluation of the service, enabling co-creation with the participation of users, creators, authors, and other stakeholders immersed in the service environment. Our study follows the criteria of the Mixed Methods Article Reporting Standards (MMARS) (See S1 Fig). The study was conducted between July and November 2023.

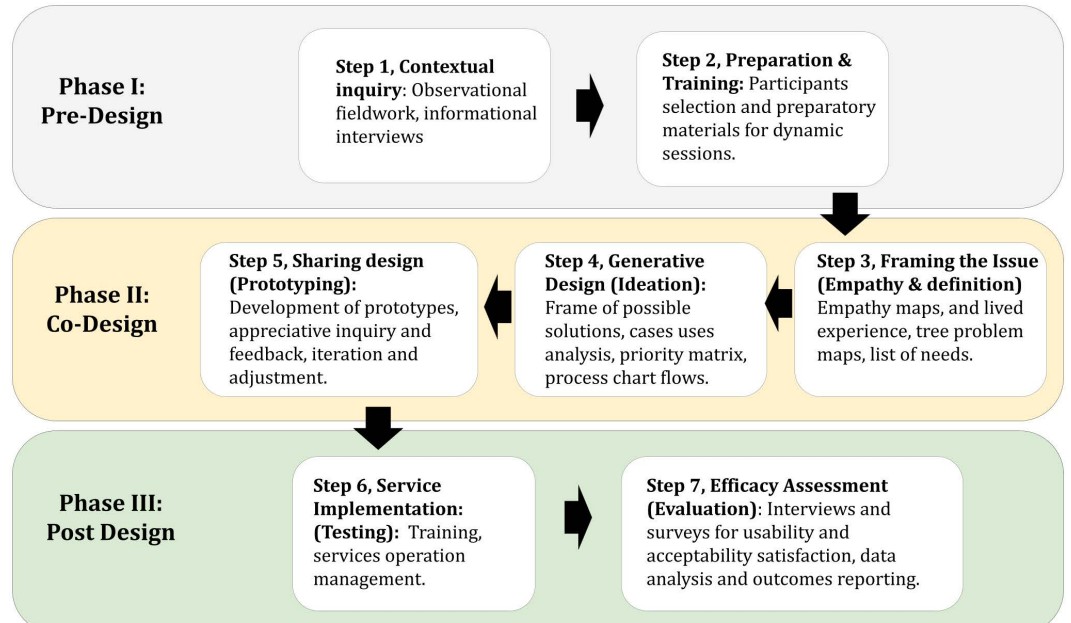

**Fig 1. Phases of the Methodology Design for TeleHealth Service Aimed at Identifying Mental Health Issues in Vulnerable Occupational Groups in Peru.**

## Phase I: pre-design

This phase had 1 month and half duration; a bibliographic review of regulations, backgrounds and best practices for implementing telehealth services targeting prioritized occupational groups was conducted for the contextual inquiry analysis (step 1), enriching the requirements for the development of the telehealth platform and the service implementation. Furthermore, a qualitative approach was employed, using semi-structured interviews to explore the needs, barriers, and limitations faced by users participating in telehealth services for mental health.

For one month, 23 volunteers with a history of COVID-19 were interviewed and participated (five teachers, five healthcare professionals, five police officers, five healthcare professionals responsible for telehealth, and three health decision-makers). Fifteen interviews were conducted with health professionals, teachers, and police officers. Inclusion criteria required participants to score high on depressive or anxious symptoms, as assessed by the PHQ-2 and GAD-2 tools. These evaluations were part of the occupational activities conducted by CENSOPAS-INS (National Center for Occupational Health and Environmental Protection, in Spanish). Police officers were invited by DIRSAPOL (Police Health Directorate, in Spanish) and selected using intentional non-probabilistic sampling. Teachers were invited from UGEL 3 (Local Educational Management Unit, in Spanish). Additionally, five interviews were conducted with healthcare professionals, primarily psychologists or physicians involved in teleconsultations, who provide mental health care in Community Mental Health Centers in Lima or other Ministry of Health (MINSA) facilities. These professionals were sourced from centers such as San Gabriel Alto, La Perla, and Carabayllo. Three interviews were also conducted with decision-makers, defined as public officials responsible for administrative, managerial, or decision-making roles in telemental health. These decision-makers came from institutions including the Directorate of Mental Health (DSAME) within MINSA, the General Directorate of Telehealth, Emergencies, and Referrals (DIGTEL), and EsSalud's National Telehealth Center (CENATE). They were invited through formal letters issued by the Directorate of CENSOPAS-INS.

An anthropologist and the research team conducted the interviews, lasting 20–30 minutes each. The informed consent was read before the interview. Audio recordings were made and later transcribed for analysis. The transcripts were then segmented by topics and subtopics according to the category of users (external and internal). Following data collection, information was coded and analyzed using a thematic analysis with the assistance of Atlas Ti [45]. The information obtained was anonymized to ensure the protection of participants' identity data during the analysis.

Preparation materials and training resources (step 2) were required to prepare for the next co-design sessions. In this step all participants were selected for the next phase.

### Phase II: co-design

This phase lasted for two months and involved the development of the service. It involved 12 participants, four psychologists with extensive experience in telehealth, one software engineer, one health professional specialist in software development, and the 6 members of the research team. The professionals were hired based on the selection of résumés and experience, and the research team was composed of the authors of this article. Together, they delineated the design characteristics of the care processes and the development of the telehealth platform prototype through co-creation workshops based on the application of Design Thinking methodology (*Emphaty, definition, ideation and prototype design*) [34].

Drawing from the information gathered in Phase 1, co-design tools such as empathy mapping, lived experience discussion, problem tree maps, and list of needs were employed by the participants throughout workshops for framing issues cases (step 3). For empathy mapping, participants in a workshop were asked to think like users and place stickers with key ideas related to what users say, think, do, or feel on a paper banner. The live experience discussion consisted of a focus group workshop where participants shared opinions and personal experiences as users. The problem tree mapping was a dynamic activity where participants collaboratively built a tree representing the causes and consequences of the main problems faced by users. This information was contrasted and consolidated into key problems, causes, consequences, feelings, and user mindsets, alongside the knowledge gathered from the interviews conducted in Phase I. The results helped to develop the next dynamics of the co-design workshops.

In the generative design activities (step 4), participants developed a framework of possible solutions, using brainstorming maps to explore available solutions for the identified problems and causes. The insights from the previous dynamics were organized into a creative matrix, then further arranged and evaluated in an impact matrix. As a result of the workshops, the most valuable insights were used to define prototype models.

Information compiled from these sessions served to delineate the functional requirements of the platform prototype alongside service care guidelines. The information obtained from the group workshops was documented through flowchart models and functional requirement lists, which helped the development team design the platform prototype. No further data processing of participants was required, as the data was anonymous. Following this, the platform web prototype was developed by the software specialists' team and shared with the team for testing (step 5). First, it stemmed from drawings, image mock-ups, Excel spreadsheet templates and Google Forms tools that evolved towards a functional and integrated web platform through a three-iteration design process. It was possible with the participation of tele-psychologist professionals and the research team that validated each stage of platform development. The early use of no-code tools from the prior stages of web development facilitated the definition of the service model process, the structuring of care forms, automated calculation algorithms, response listings, registration flow automation, and care status management (See S5 Fig).

Ultimately, pilot tests of the service were conducted, preceded by virtual training sessions and direct observation usability evaluations with the participants and some early users. Service feedback facilitated three iterations of adjustments to the initial proposal, culminating in the final version of the technological proposal for use in the subsequent phase [46–48].

## Phase III: post-design

It spanned three months of intervention and evaluation of the telehealth service designed during Phase 2 (Service care guidelines and the functional prototype of the telehealth platform). The deployment comprised two sub-stages:

**Telehealth service intervention.** Three tele-psychologists and one service operator were trained in using the model service guide. These professionals were hired specifically for this operation. Then, they carried out operation management of telehealth service under researchers' supervision (step 6). Among the primary activities, coordination with healthcare organizations, educational centers, and the national police sector was conducted to promote the screening service to the target audience. A social media communication campaign was also deployed to pre-identify potential participants. Then, pre-identified users were approached by a service operator to schedule a psychoeducation session. The service concluded with issuing the final report, a certificate of attendance, and guidance for the timely referral, just in cases that required specialized clinical treatment [49].

Several resources were used to monitor the efficiency of care attention, such as Google Analytics for tracking web user metrics and Google Looker Studio for assessing care progress [50].

The custody and processing of personal and sensitive data during the service deployment were controlled by the service operator. Access levels were established between the operator and the tele-psychologists for the use of patient data from the service. During the results analysis process, the data were anonymized to protect the participants.

**Usability, satisfaction and acceptability assessment.** The telehealth platform's efficacy was assessed by quantitative and qualitative instruments (Step 7). A quantitative evaluation for usability assessment was applied by online survey application to 81 patients who utilized the service. The patients were pre-identified as at risk based on a mental health questionnaire and had participated in at least one psychoeducation session. They consisted of healthcare workers, teachers, and police officers. It employed the "Computer System Usability Questionnaire Scale (CSUQ version 3)", which assessed three dimensions: system quality, information quality, and interface quality [51,52].

The qualitative evaluation for acceptability assessment comprised semi-structured interviews directed towards six external users (patients), three of whom received at least two service components, three who received only one component, and three internal users (operators and tele-psychologists) [45]. On the other hand, satisfaction was evaluated through two methods, qualitatively by the semi-structured interviews and quantitatively by the "Outpatient Medical Consultation Users´ Satisfaction Scale (ESCOMA)" applied to 81 patients, assessing satisfaction across two dimensions: satisfaction with care management processes and satisfaction with psychological care [53]. The information from the surveys was analyzed anonymously, usability scores were calculated for the quantitative data, and participants' opinions were categorized by topics and subtopics for the qualitative data. All the quantitative information was structured and analyzed using SPSS 23.0 and qualitative information was coded and analyzed using a thematic analysis with the assistance of Atlas Ti.

## Ethical considerations

In general, the study provided online informed consent to all participants, who took part voluntarily. Informed consent was signed virtually by participants by clicking on the "I agree to participate" option. Additionally, the identity of participants in interviews, workshops, and surveys was protected through data anonymization and the custody of information with controlled access levels. Participants did not receive any type of monetary contribution for their participation.

This study was approved by the ethics and research committee of the National Institute of Health of Peru, through Directorial Resolution No. 206–2023-OGITT-INS (COD. OC-048–22).

## Results

### Phase I: pre-design

The main outcomes of the initial phase, which involved interviews with 23 users and stakeholders in the mental telehealth ecosystem services were identified, highlighting common opinions. These included principal topics as: a) Dissatisfaction

with telehealth services was perceived as "very impersonal" and "distant". Also, lacking the use of clinical records; b) Issues with scheduling and appointment times related to flexible schedules and missed appointments; c) Training for healthcare workers in telehealth described as issues with rapport between patients and healthcare professionals and connectivity problems; d) The need for healthcare system integration, described as a lack of infrastructure and the need for more coordination with other governmental entities.

## Phase II: co-design

**Characteristics of telehealth service:.** The telehealth service model was named TelePsico CENSOPAS and comprised four general processes: a) Service promotion; b) User pre-identification; c) Appointment management; d) Psychoeducation counseling and referral.

The service promotion activities aimed to attract potential users through talks and educational training on mental health in the workplace with allied institutions and the dissemination of advertising about the service's benefits on social networks. Subsequently, users accessed the TelePsico web platform where they could find information about the service and access educational information.

Next, users accessed the mental health risk pre-identification evaluation questionnaire through the telehealth website. The form included psychometric assessment instruments: Patient Health Questionnaire PHQ-9 for detecting depressive symptoms, Generalized Anxiety Disorder GAD-7 for detecting anxiety symptoms, and the Posttraumatic Stress Disorder Checklist for DSM-5 (PCL-5) for detecting posttraumatic stress. Demographic, contact and sociodemographic data were also obtained. Upon completing the form, users received an email with an automated personalized message about their evaluation results, and those at risk were invited to schedule a psychoeducation session.

Appointment scheduling was managed by a service operator who downloaded the data of pre-identified users to send private communication through a project-exclusive WhatsApp channel. In coordination with the users, scheduling times for appointments were established. Additionally, the channel was used for reminders and communication support. Scheduled appointments were recorded in the TelePsico web platform for telehealth management, so the attending psychologists could address them.

Psychological counseling and referral consisted of a video call session between the user and a psychologist, lasting 40 minutes for an interview. The session allowed for the consolidation of the patient's mental health diagnosis. It was recorded through an assessment report, proof of care, and guidance, which was subsequently delivered to the patient via email after the interview for timely referral if necessary.

The characteristics of the service are described in four steps (See Fig 2).

**Characteristics of telehealth platform design.** The development of the telehealth platform required integrating various digital technologies, including an owner development platform and third-party service technologies that allowed communication support and information exchange.

The prototype development was carried out iteratively through three development cycles. In the first cycle, flowcharts of the processes for the digital tool were created. These were then drawn as mock-ups and structured in Excel tables to define the records and database structure, along with the use of Google Forms templates for the initial forms.

In the second cycle, this early prototype was tested with users during the co-design process until it was validated and finalized. No-code tools from Google were then used to design and define the prototypes, as well as to structure the care forms, automate calculation algorithms, manage response listings, automate registration flow, and handle care status management (See S5 Fig).

In the third cycle, the proprietary telehealth platform prototype developed by the research team was named the TelePsico Web Platform (https://telepsicocensopas.ins.gob.pe/) (See S6 Fig). Additionally, third-party service platforms were used as complementary technology tools in the service flow. These included video calling platforms like Zoom and Google

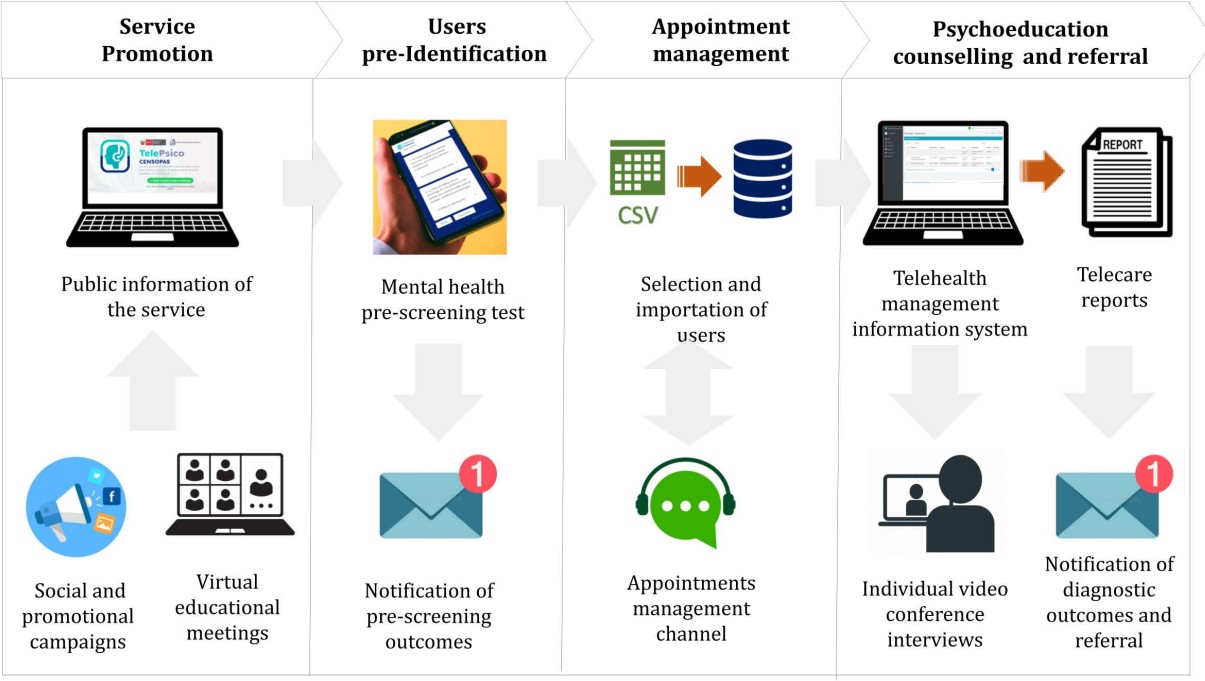

**Fig 2. Telehealth Service Delivery Process Flowchart Aimed at Identifying Mental Health Issues in Vulnerable Occupational Groups in Peru.**

Meet, communication tools such as WhatsApp, and Google Sites for designing the information website, and social networks like Facebook, LinkedIn, and email (Google Gmail).

These three development cycles, from basic prototypes to more complex models, were carried out by considering all the possible functional components that had been previously agreed upon in the workshops. Unlike incremental development, where new functionalities are added to the final prototype according to user needs, this development method allowed for saving investment resources and development time. The description of the digital modules used is detailed in Table 1.

### Phase III: post-design

**Telehealth service intervention outcomes:.** The pilot test deployment reached 3900 users who visited the informational website. Among them, 698 users completed the mental health screening questionnaire, pre-identifying 193 patients with symptomatology of risk (see S4 Fig). Of these, 134 users attended psychoeducation sessions. Finally, 81 patients responded to the phase 3 evaluation surveys (satisfaction and usability).

External users (potential patients) were mainly composed of women 527 (76%). Forty percent of the participants were between 25 and 35 years old, and 2% were over 65 years. Regarding occupation, 85% were healthcare workers, followed by the education sector (14%). Additionally, participants were asked if they had received any previous diagnosis in mental health before their participation; 8% did before the pandemic, and 4% did during the pandemic. The characteristics of the 698 participants who completed the screening test surveys are shown in Table 2.

One hundred ninety-three individuals (27.7%) were pre-identified with some mental health symptomatology. Fig 3 describes the distribution of workers with single or associated risk symptoms. Among them, mainly 149 workers exhibited symptoms of post-traumatic stress. Workers' symptoms distribution by job position are described in S7 Fig.

**Table 1. Telehealth software functionality requirements for service flow in identifying mental health issues in vulnerable occupational groups in Peru.**

| Requirements | Functionality description | Inserted into TelePsico web platform | Digital application resources |
|---|---|---|---|
| Public information of the service | A responsive website tailored for web browsers on PCs and mobile devices, displayed general service information, as well as announcements for talks, explanatory videos, and psychoeducational materials for patients and the general public. | Yes | TelePsico web platform |
| Social and promotional campaigns | Social network profiles serve as communication channels with other users, enabling the sending of educational messages and other means of promoting the service. Additionally, links to free courses and mental health conferences were disseminated through the WhatsApp channel and social media platforms, aiming to provide knowledge and preventive tools in mental health to the target population and encourage enrollment in the TelePsico service. | No | WhatsApp channels/ Facebook pages/ LinkedIn pages |
| Virtual educational meetings | Video calling platform for conducting educational meetings with allied institutions and communities of occupational health and safety workers. | No | Google Meet/ Zoom |
| Mental health pre-screening test | Web-responsive form questionnaire integrated into a public web information site. This resource calculated pre-screening outcomes for mental health risk using algorithms for the PHQ-9, GAD-7, and PCL-5 instruments. | Yes | TelePsico web platform |
| Notification of pre-screening outcomes | The system allowed integration of the screening test registration form with automated email responses through service integration, such as Google AppScript, enabling personalized messages to be sent regarding questionnaire results. Messages were tailored based on whether users were at risk or not for mental health issues, as determined by the results of the depression, anxiety, and post-traumatic stress questionnaires. | Yes | Google App Scripts and Gmail integration |
| Selection and importation of users | Web tool for downloading contacts of pre-identified users. It allows the importation in CSV format of selected contacts for scheduling appointments in the system. | Yes | TelePsico web platform |
| Appointments management channel | An exclusive WhatsApp user profile was assigned for the service for communication with potential users and operator service. | No | WhatsApp contact |
| Telehealth management information system | Intranet module designed for internal users of the service (researchers, service operators, and tele-psychologists) in a web format, allowing for appointment scheduling and management of the status processes of care (initiated, in progress, attended, completed, and notified) for service progress monitoring. The system had different access levels among users, ensuring the protection of patients' personal and health data. | Yes | TelePsico web platform |
| Individual video conference interviews | Video calling platform for conducting individual psychoeducational sessions between psychologists and users. | No | Google Meet/ Zoom |
| Telehealth reports | The system enabled tele-psychologists to fill out psychological medical records for the generation of reports and certificates of care. Researchers could access progress reports on care through integrated dashboards. | Yes | TelePsico web platform Google Analytics Google Looker Studio |
| Notification of diagnostic outcomes and referral | An exclusive email account for the project was utilized, managed by the service operator, for sending individual reports and certificates of care following the appointment, as well as providing guidance for timely referral if required by the diagnosis. | No | Google Gmail |

**Usability evaluation outcomes.** Concerning quantitative evaluation analysis; 81 external users (patients) who participated in psychoeducation sessions and had identified risk symptoms were surveyed. As a result, an average score of 86.1 ± 16.9 SD was obtained on the Computer System Usability Questionnaire Scale (CSUQ version 3), which is classified at an "Excellent" level within the SUS scale. The following scores were observed for usability dimensions: D1 - System Quality 38.0 ± 5.6, D2 - Information Quality 36.3 ± 6.9, and D3 - Interface Quality in external users (patients) 24.3 ± 5.1 (See S5 Fig); Of the 81 external users, 25 (30.9%) rated the usability of the platform as "Best imaginable",

**Table 2. General characteristics of external users who completed the screening test from TelePsico service.(N = 698).**

| Sociodemographic characteristics | Participants |
|---|---|
| **Gender, n (%)** | |
| Female | 527 (76) |
| Male | 171 (24) |
| **Age group (years), n (%)** | |
| <25 | 46 (7) |
| 25-35 | 280 (40) |
| 36-45 | 223 (31) |
| 46-55 | 90 (13) |
| 56-65 | 48 (7) |
| >65 | 11 (2) |
| **Occupational group, n (%)** | |
| *Education workers* | **97 (14)** |
| Teacher | 50 (7) |
| Education administrative staff | 47 (7) |
| *Police workers* | **9 (1)** |
| Police officers | 5 (1) |
| Police administrative staff | 4 (1) |
| *Health workers* | **592 (85)** |
| Healthcare Workers | 439 (63) |
| Health administrative staff | 153 (22) |
| **Work Modality, n (%)** | |
| On-site work | 606 (87) |
| Remote work | 19 (3) |
| Hybrid work | 73 (10) |
| **Diagnosis of mental health, n (%)** | |
| No | 613 (88) |
| Yes, before the pandemic | 57 (8) |
| Yes, during the pandemic | 28 (4) |

followed by 26 users (32.1%) who rated it as "Excellent", 15 users (18.5%) "Good", and 10 users (12.3%) as "Okay". Conversely, lower ratings included 3 users (3.7%) who rated the usability as "Poor" and 2 users (2.5%) as "Horrible" (See S6 Fig).

The qualitative evaluation analyzed various topics from responses described by external users during the interviews. They expressed quick adaptation, ease of system use, and understanding of instructions. Additionally, they suggested the need to incorporate detailed instructions for individuals with limited knowledge of using digital technologies, incorporating blog-style informative components and increased promotion on social media platforms. On the other hand, internal users expressed rapid learning of tool usage and satisfaction with saving time in patient report development using the platform. However, they emphasized simplifying the registration system and promoting platform usage training for all users (See S7 Fig).

To ensure the reliability of the data, the quantitative surveys were conducted online through Google Forms, which were administered to the personal email accounts of the participants. The qualitative surveys were also transcribed and classified by a specialist assigned to this task.

**Service satisfaction and acceptability evaluation outcomes.** Eighty-one external users (patients) were surveyed for assessing service satisfaction. The survey results revealed an 82.7% (67 patients) medium-high satisfaction rate

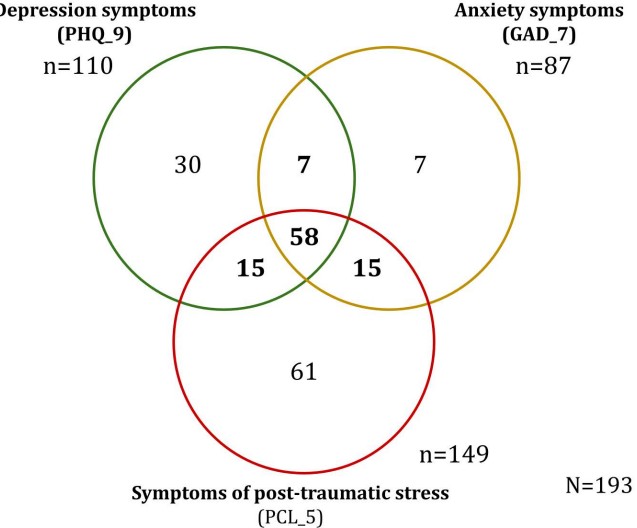

**Fig 3. Distribution of pre-identified workers from Vulnerable Occupational Groups in Peru with symptoms of mental health risk admitted to the TelePsico service.**

for the "Satisfaction with care processes" dimension and a 91.4% (74 patients) medium-high satisfaction rate for the "satisfaction with psychological care" dimension (See S8 Fig). Overall, external users exhibited higher satisfaction with the psychological care received than care management processes, including service provision, appointment scheduling, and result notification processes.

The acceptance evaluation was conducted through in-depth qualitative semi-structured interviews with both external and internal service users. Six internal users (patients) and three external users (telepsychologists and service operators) were interviewed. The categories of acceptability evaluation responses in external and internal users are described in S9 Fig.

Regarding the perception of acceptability among external users, they expressed a positive experience using the service, highlighting the ease of access for telehealth through any device. Another aspect mentioned was the trust in the service generated through dissemination activities such as educational talks and sharing information within professional groups, which instilled a sense of security in users regarding the service's credibility. Areas for improvement include the need to standardize messages from the operator, and enhance the efficiency of meeting scheduled appointments. They also suggest the creation of support communities and social media outreach.

On the other hand, internal users reported an easy adaptation to using the platform during the service, especially in accessing information about scheduled patients and the quick preparation of reports, which contributed to their service's efficiency. Opportunities for improvement include optimizing the user experience in the system, providing training to the target audience to promote the service, and offering education on digital literacy for users with limited digital skills.

## Discussion

This study presents the development and evaluation of a telehealth service model tailored for vulnerable occupational groups, named TelePsico CENSOPAS. The model comprises four key processes: service promotion, user pre-identification, appointment management, and psychoeducation counseling and referral. The service demonstrated high usability and user satisfaction. These findings indicate that the TelePsico model is well-accepted among users, particularly

in structured occupational health environments, compared to traditional outpatient services in Peru. This study underscores the potential of user-centered design frameworks, such as Design Thinking, to create effective telehealth solutions within the occupational health sector.

## Barriers and limitations to the development of a telehealth service

During the pre-design phase, we identified users and other stakeholders within the telehealth service ecosystem and characterized their needs, limitations, and barriers. The main barriers included uncertainty in appointment management, non-compliance with scheduled service times, lack of clear communication, and the absence of a legal framework that integrates care outcomes with the healthcare system to enable timely referral. Although Peru has made significant improvements in its regulatory framework, allowing the development of multiple telehealth services, legal and operational gaps persist. These gaps limit sustainable satisfaction and adherence and hinder the integration of telehealth services into the broader healthcare system [54], particularly when services are not developed with end users in mind or fail to address digital divides [55]. Rees and Peralta note that many telehealth services prioritize solving internal operational problems within healthcare systems without adequately considering patient needs, which negatively affects sustainability and acceptance [44]. Given this reality, it is proposed to follow Bird's recommendations [35] for user-centered development, which was employed in the present study, to ensure its replicability and sustainability. In response to limitations identified during interviews with users and decision-makers, various solutions were implemented in the service's development, focusing on software structure and care procedures. A system was established to allow access via a link to a pre-screening mental health test, ensuring immediate entry to the initial service stage. If risk indicators are present, direct communication via WhatsApp is initiated to set the appointment date, serving as both a reminder and a means for clear communication. The TelePsico web platform integrates clinical history management, report generation, and psychological assessments while facilitating access to virtual rooms for telehealth services. Additionally, it provides a channel for disseminating service information and mental health care recommendations in the workplace. For more details, see Table 1.

## User-Centered design and user feedback approach

The co-design phase activities allowed researchers and project stakeholders to develop a service model and platform in an agile, participatory, and iterative manner. Methodologies such as Design Thinking have been widely used in recent years in developing innovative telehealth services due to their user-centered approach [34,56]. In this sense the co-design of minimum viable products (MVPs), has proven to be cost-effective and enable cost reduction in resources and maintenance, increasing productivity, acceptance, and effectiveness, particularly in healthcare settings with limited resources and high resistance to change [55,57,58].

The pilot implementation of the solution during the post-design phase (Phase 3) reached 698 users through coordination with occupational safety and health services, as well as human resources departments of entities corresponding to the target occupational groups. Additionally, educational talks on mental health were conducted, and digital marketing strategies, the creation of official communication channels, and a website were utilized. These actions elicited recommendations among users and professional circles, facilitating service satisfaction and usability feedback. In this regard, Otto L. et al. developed a working model called the Telemedicine Community Readiness Model (TCRM), which emphasizes the need to involve patient communities from prototype development to advanced evaluation stages. This model contributed to users not only being passive recipients but also adopting an active role as part of patient communities with common interests capable of providing solutions during service validation [59]. While previous studies highlight the importance of user-centered design in telehealth development, our study extends these findings by demonstrating its applicability in occupational health settings within a middle-income country context.

 

## Healthcare workers, teachers, and police officers are the most vulnerable

During the post-design stage, the pilot aimed to enhance access to mental health services among the target occupational groups. To achieve this aim, coordination was established with various institutions, such as healthcare networks, educational authorities, and law enforcement agencies. A higher participation rate was observed among healthcare workers (85%) than those in the education and law enforcement sectors. One possible reason is the working conditions experienced by other occupational groups as teachers and police officers, which might have limited their access to the service, such as conflicting work schedules that could have interfered with outreach activities. The commitment from occupational safety and health services facilitated user participation and trust, significantly promoting mental health and digital health education in the workplace. These aspects should be strengthened beforehand in the target audience to ensure adherence and acceptability of telehealth services [60–62].

Regarding the screening results, the high frequency of post-traumatic stress symptoms, anxiety, and depression among healthcare professionals is noteworthy. These findings align, particularly with what has been reported by other medical professionals as nurses in the context of the COVID-19 pandemic, given the nature of their work (caring for critical patients, frontline work, etc.) [63]. Even in Peru, results have reported that 21.7% healthcare workers in Peru experienced severe anxiety, whereas 26.1% of them experienced severe mental distress [64,65]. Meanwhile, authors such as Poug-net et al. suggest that this prevalence was high even before the pandemic [66]. On the other hand, during the pandemic, workers in the education sector faced significant psychosocial challenges, such as fulfilling class requirements and working long hours remotely. Similarly, chronic work-related stress is identified as one of the most significant occupational risks affecting the mental health of police officers [67]. Despite this, there is insufficient information about the mental health of teachers and police officers in Peru. Therefore, it is necessary to promote programs and policies for mental health surveillance in the workplace for these occupational groups [68].

## Disruptive organisations for user satisfaction

Regarding the telehealth service, the proposed model aimed to be disruptive regarding organizational structure and operational management compared to the traditional model. The functional structure was based on the model proposed by the US Department of Health and Human Services, which included various roles such as general coordination, supervision, training, and device management by the team [49]. Telehealth service providers (psychologists) and scheduling managers (operators) provided flexibility in service delivery due to the demand for appointments outside of regular hours, facilitating communication and user participation. These benefits have been previously discussed by Sanders et al. [69], who reinforce the principles of communication, development of tracking metrics, and evaluation of progress in care for the rapid control, assessment, and adjustment of service distribution strategies. Additionally, according to the Framework for the Implementation of a Telemedicine Service recommended by the WHO, organizations need to redesign their human resource management, processes, and working conditions in new scenarios, such as telemedicine, as the effectiveness of implementing these services is evaluated [70].

During the evaluation of the usability of the software platform, quantitative and qualitative techniques were combined, including the System Usability Scale (SUS) survey, semi-structured interviews, and direct observation [47,48,71]. These techniques are widely used in telehealth project development and help obtain a broader understanding of the service experience. It is worth noting that the evaluation process was ongoing throughout the development, even from Phase 2, where usability was assessed through direct observation with internal users during initial tests. As a result of the pilot, a usability score of 86.1 ± 16.9 SD was achieved, qualifying it as a product with a high level of acceptance among users. Authors like Hyzy et al. describe a usability level above 68 ± 12.5 SD as acceptable for software and hardware products, also applicable in the healthcare technological solutions field. Additionally, the authors describe in their review study an average usability score of 76.64 ± 15.12 SD for computer applications in healthcare [72]. In comparison with previously reported usability scores for healthcare applications, which average

around 76 points, the higher SUS score observed in our study suggests that the iterative co-design process may have contributed to enhanced user acceptance.

## Facilitators of the platform and service

As the main facilitator of service usage, external users identified the simplicity of the service usage on the platform, which required few steps for scheduling, while internal users highlighted the management of care and the rapid generation of care reports. On the other hand, both user groups described opportunities for improvement in the operator-patient communication flow, use of standardized messages, and active use of social networks for service positioning in user communities, as well as simplifying the system usage experience. At a functional level, one of the features that most contributed to users having a high degree of usability was the integration of third-party application services such as WhatsApp, video calling systems like Meet, Zoom, and Google Calendar, which were already in everyday use among users. Additionally, the use of a multi-device web system allowed easy accessibility on mobile phones and PCs. Presently, the increasing use of API-based technologies (Application Programming Interfaces) allows interoperability between various systems, streamlining development processes. However, the current debate on their use revolves around information security [73,74]. It is necessary to mention that participants' health data were not shared by third-party services, and the tools used only facilitated the operational management of service distribution.

The level of service usage satisfaction, evaluated using the ESCOMA instrument, showed a high score in satisfaction with care processes and psychological care. The evaluation instrument was initially developed by Moscoso et al., who found lower scores in satisfaction levels among users of MINSA, EsSalud, the Armed Forces, and private entities who received outpatient medical care services. These differences could be attributed to system characteristics, such as the freedom of scheduling appointments and the accessibility described earlier, which meet users' needs and expectations [53]. However, it should be noted that the study by Moscoso et al. was conducted at the national level in in-person settings before the COVID-19 pandemic, whereas our study was conducted in telehealth settings during the COVID-19 pandemic. Therefore, the studies may not be directly comparable. Despite these contextual differences, the comparison provides a useful reference point suggesting that telehealth-based models may achieve higher satisfaction levels than traditional outpatient care under certain conditions.

Additionally, a higher satisfaction score was identified regarding psychological care than the care management process. One possible reason could be that, during the pilot development, adjustments had to be made to the service management processes, so early users may have yet to experience high satisfaction with the operational management of appointments and result delivery.

Furthermore, age and previous experience with similar services facilitated a quick adaptation to using the service. Users considered transparency, accuracy of information, professional assurance, and appropriate handling of personal health data to be important requirements for service acceptance. The use of digital devices during telehealth operations requires the responsible handling of personal and sensitive patient data. Some of these mechanisms include verifying patient identity during teleconsultations, encrypting and securely storing access data and sensitive information. It is essential to consider what type of information is shared with third parties when using complementary services, ensuring access levels are controlled, as well as providing training and digital literacy for the team of professionals delivering the service [24].

These aspects align with the concept of user-centered security developed by Vega-Barbas et al [75]. Among other acceptance qualities, the structured flow of care, use of validated mental health assessment tools, psychoeducation service provided by accredited psychologists, and institutional prestige contributed to greater participant and community confidence in using the service. All these characteristics ultimately shaped the service's business model, many of which coincide with the model for an online assistance service proposed by Van D. et al [46].

## Implications for digital mental health

For digital interventions initially developed in controlled research settings to be effectively implemented in real-world contexts, they must be supported by adequate digital infrastructure, engage relevant stakeholders, and be embedded within health systems through enabling policies [38,76]. However, cultural adaptation and personalization are crucial to successful implementation, especially in low- and middle-income countries. This study offers a framework that can be applied and assessed in similar contexts, highlighting the importance of a user-centered model. Such a model should integrate tools from implementation science and evidence-based insights gathered from users throughout the implementation process. This approach can enhance the impact and sustainability of digital health interventions by ensuring they are both relevant and adaptable to diverse user needs [77].

## Strengths and limitations

The present study makes a valuable contribution to the field of telehealth and its development; however, several significant limitations are identified. The project's development time, spanning six months, proved challenging due to the high demand for design activities and solution iteration, leading to an accelerated process in the final phase, achieving three iterations. It is advisable to conduct more iterations before implementing the service in healthcare facilities to tailor it to their needs [35,46,78]. Wachtler et al., in their study on the design of an app for early detection of depression, also developed their proposal in three iteration cycles [79].

Furthermore, the participation of occupational groups, such as the educational and police sectors, was limited due to the need for more organization in their respective workplaces' occupational health and safety services, highlighting the need for a more robust institutional commitment. In terms of integration with the healthcare system, the lack of updating of the Peruvian legal framework and the insufficient availability of mental health services pose obstacles to the effective referral of users diagnosed with conditions requiring specialized psychological or psychiatric care [23]. Therefore, it is necessary to conduct more studies and surveillance programs that address the early identification of psychosocial risks in the work environment of these occupational groups, as well as protection and intervention measures. Additionally, integrating occupational health and safety services with national health systems is a need that can be addressed not only from a regulatory standpoint but also through the interoperability of information systems for their monitoring.

On the other hand, the use of the ESCOMA instrument for satisfaction evaluation, while showing comparatively promising results for the telehealth proposal of the study, could not be compared with other telehealth services due to the novelty of the instrument.

Despite these limitations, the study offers a holistic approach to product design, focusing on the user and combining qualitative and quantitative components in its methodology. The use of this framework, which involves the co-design of products and services through development phases, allows organizations to establish a structured and efficient work plan regarding the use of investment resources, time, and implementation in an orderly and systematic manner. The experience presented in this research can be replicated in the field of health services, especially in the context of occupational health services [35,78]. Additionally, it prioritizes the attention to vulnerable occupational groups in the context of health emergencies, such as healthcare workers, teachers, and police officers, who face severe limitations in accessing mental health services, particularly during the COVID-19 pandemic [80,81].

Future research related to the design and implementation of new technologies in the field of health services in the Peruvian context will provide further evidence to replicate these user-centered design methodologies across different levels of organizations in Peru. It is necessary to replicate these experiences to increase the impact of implementation at the national health services level. Furthermore, the possibility of incorporating tools such as Chatbots operated with Artificial Intelligence in future projects is proposed to enhance the operational efficiency of telehealth services and complement the work of healthcare professionals [82–85].

## Conclusion

In conclusion, the study enabled the development of a tele-mental health service and platform for the early identification of mental health risks in vulnerable occupational groups such as healthcare workers, teachers, and police officers. To achieve this, user-centered design methodologies were employed, using co-creation techniques with the participation of users themselves during three phases of development (formative, co-design, and implementation). Optimal satisfaction was achieved in the management of care processes and the psychological care provided, compared to outpatient services in Peru (MINSA, Armed Forces, National Police, and private). Additionally, the tele-mental health platform achieved a usability score, which rated the product with a high level of acceptance among users, surpassing the minimum acceptability required for software and hardware products, and the average for computer applications in healthcare.

## Supporting information

**S1 Fig. Mixed Methods Article Reporting Standards.**
(DOCX)

**S2 Fig. Early prototypes in co-design phase.**
(DOCX)

**S3 Fig. TelePsico web platform prototype.**
(DOCX)

**S4 Fig. Distribution of workers identified with mental health risk symptoms by job position.**
(DOCX)

**S5 Fig. Distribution of satisfaction and usability scores in external users (ESCOMA and SUS) according to sociodemographic characteristics.**
(DOCX)

**S6 Fig. Distribution of usability rating in external users (SUS) according to sociodemographic characteristics.**
(DOCX)

**S7 Fig. User Perceptions during Usability Evaluation through Interviews.**
(DOCX)

**S8 Fig. Service satisfaction qualification levels by ESCOMA instrument D1 and D3 dimensions.**
(DOCX)

**S9 Fig. Perceptions of external and internal users regarding the acceptability of the service.**
(DOCX)

## Author contributions

**Conceptualization:** Jimmy Andreyvan Cainamarks-Alejandro, Miguel Angel Burgos-Flores, Jonh Astete-Cornejo.

**Data curation:** Jimmy Andreyvan Cainamarks-Alejandro.

**Formal analysis:** Jimmy Andreyvan Cainamarks-Alejandro, Miguel Angel Burgos-Flores.

**Funding acquisition:** Jonh Astete-Cornejo.

**Investigation:** Jimmy Andreyvan Cainamarks-Alejandro, Liliana Cruz-Ausejo, Miguel Angel Burgos-Flores, Jaime Rosales-Rimache, Jonh Astete-Cornejo, David Villarreal-Zegarra.

**Methodology:** Liliana Cruz-Ausejo, Miguel Angel Burgos-Flores, Jaime Rosales-Rimache.

**Project administration:** Miguel Angel Burgos-Flores, Jonh Astete-Cornejo.

**Resources:** David Villarreal-Zegarra.

**Software:** David Villarreal-Zegarra.

**Supervision:** Jimmy Andreyvan Cainamarks-Alejandro, Jonh Astete-Cornejo, David Villarreal-Zegarra.

**Validation:** Jimmy Andreyvan Cainamarks-Alejandro, Liliana Cruz-Ausejo, Miguel Angel Burgos-Flores, Jaime Rosales-Rimache, David Villarreal-Zegarra.

**Visualization:** Liliana Cruz-Ausejo, Miguel Angel Burgos-Flores.

**Writing – original draft:** Jimmy Andreyvan Cainamarks-Alejandro, Miguel Angel Burgos-Flores.

**Writing – review & editing:** Jimmy Andreyvan Cainamarks-Alejandro, Liliana Cruz-Ausejo, Miguel Angel Burgos-Flores, Jaime Rosales-Rimache, Jonh Astete-Cornejo, David Villarreal-Zegarra.

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
