## [Decision Letter · Decision Letter 0]

11 Dec 2025

Dear Dr. Villarreal-Zegarra,

Thank you for submitting your manuscript to PLOS ONE. After careful consideration, we feel that it has merit but does not fully meet PLOS ONE’s publication criteria as it currently stands. Therefore, we invite you to submit a revised version of the manuscript that addresses the points raised during the review process.

We look forward to receiving your revised manuscript.

Kind regards,

Jahanpour Alipour, Ph.D.

Academic Editor

PLOS One

Journal Requirements:

We express our gratitude to the Spanish Agency for International Development Cooperation (AECID, in spanish) and the National Program for Scientific Research and Advanced Studies (PROCIENCIA-CONCYTEC-Peru, in spanish), who funded the project, enabling its execution, under contract number 076-2021-PROCIENCIA

5. Please amend your list of authors on the manuscript to ensure that each author is linked to an affiliation. Authors’ affiliations should reflect the institution where the work was done (if authors moved subsequently, you can also list the new affiliation stating “current affiliation:….” as necessary).

6. Please remove your figures from within your manuscript file, leaving only the individual TIFF/EPS image files, uploaded separately. These will be automatically included in the reviewers’ PDF**.**

7. We note that Figure 1, Supplementary A, Supplementary B, Supplementary C, Supplementary D, in your submission contain copyrighted images. All PLOS content is published under the Creative Commons Attribution License (CC BY 4.0), which means that the manuscript, images, and Supporting Information files will be freely available online, and any third party is permitted to access, download, copy, distribute, and use these materials in any way, even commercially, with proper attribution. For more information, see our copyright guidelines: http://journals.plos.org/plosone/s/licenses-and-copyright.

a. You may seek permission from the original copyright holder of Figure 1, Supplementary A, Supplementary B, Supplementary C, Supplementary D, to publish the content specifically under the CC BY 4.0 license.

Reviewers' comments:

Reviewer's Responses to Questions

**Comments to the Author**

1. Is the manuscript technically sound, and do the data support the conclusions?

Reviewer #1: Yes

Reviewer #2: Partly

2. Has the statistical analysis been performed appropriately and rigorously?

Reviewer #1: Yes

Reviewer #2: Yes

3. Have the authors made all data underlying the findings in their manuscript fully available?

Reviewer #1: Yes

Reviewer #2: Yes

4. Is the manuscript presented in an intelligible fashion and written in standard English?

Reviewer #1: No

Reviewer #2: Yes

Reviewer #1: Thank you for the opportunity to review this paper. The topic of a national-scale telehealth application for mental health is highly significant. The study is exceptionally well-designed and executed, demonstrating a broad scope and resulting in a practical and currently implemented system. It is commendable that the system is already operational and in use, and I have reviewed the corresponding website.

I have a few specific comments aimed at further strengthening the manuscript:

1- I recommend a thorough check of the manuscript's punctuation and grammar. For instance, in the Abstract (Background section), I noted a sentence concluding with two periods (..) instead of a single one. A general review for consistency in punctuation (commas, periods, colons, etc.) across the entire text is advised.

2- In the Introduction section, while the third paragraph makes an essential reference to prevalence data, the Introduction would be significantly improved by explicitly stating the global prevalence of mental health disorders, followed by specific prevalence statistics within Peru. The current text implies the source of the data through citations, but it should explicitly state in the text that these statistics pertain to Peru.

3- In the Method section, the description of the setting for the system's implementation is currently ambiguous. If the scope is indeed national (national scale), this should be stated explicitly and specifically in the Methods section to reflect the significant scale of the project.

4- In the Result section, the inclusion of a flowchart (Figure 2) and a view of the initial website page in the Appendix is helpful. However, to better illustrate the user experience and the functional processes outlined in Figure 2 (from screening to the final report), it is recommended to include additional screenshots of the live system/website in the main body of the paper.

5- The Discussion is well-written, thoroughly analyzing the work from various angles. To provide richer context and academic rigor, it is suggested to integrate relevant similar studies in the appropriate sections and include a comparative analysis of this work against those studies.

6- If this information is not planned for a separate publication, it would be beneficial to include some current key utilization metrics for the operational system. This could consist of the total number of individuals who have received assistance from the system to date, or other relevant indicators of current scale and impact.

Reviewer #2: This manuscript addresses a timely and significant issue regarding mental health support for vulnerable occupational groups during the COVID-19 pandemic. The application of a User-Centered Design (UCD) framework is a major strength of this study, providing a robust model for developing effective health IT solutions. However, there are specific methodological clarifications and minor corrections required before publication.

1) Methodological Terminology:In the "Usability, satisfaction and acceptability assessment" section, the analysis is described as a "phenomenological approach." Given that the results focus on usability feedback and satisfaction rather than the deep essence of lived experiences, this term seems misapplied. The analysis appears to align more closely with "Thematic Analysis" or "Qualitative Content Analysis." Please revise this terminology to accurately reflect the method used.

2) Citation Accuracy: Reference [45] (Universitat Oberta de Catalunya, Interview Toolkit) is currently cited in the context of data analysis using Atlas Ti. This reference pertains to data collection (interviews), not analysis. Please correct this citation to a source relevant to the analytical method employed.

3) Abstract Clarity: The Abstract mentions "81 participants in the efficacy assessment" without sufficient context. To avoid confusion regarding the response rate, please clarify that these 81 participants are a subset of the 134 users who received psychoeducation.

4) Comparative Analysis Context: The study compares satisfaction scores with historical outpatient data (Moscoso et al.). While valid as a benchmark, please explicitly acknowledge in the Discussion or Limitations section that comparing a telehealth service (during a pandemic) with traditional outpatient services (pre-pandemic) involves different contexts and user expectations.

5) Minor Errors and Typos: Please correct the following errors:

- Abstract: Correct the double period in "complex cases. . Telehealth".

- Abstract-Conclusion: Add a space in "groupsAdditionally".

- Ensure consistent spacing between citations and text throughout the manuscript.

**Do you want your identity to be public for this peer review?** For information about this choice, including consent withdrawal, please see our Privacy Policy

Reviewer #1: No

Reviewer #2: No

---

## [Author Response · Author response to Decision Letter 1]

2 Jan 2026

PONE-D-25-40617

Evaluation of usability and acceptability of a Peruvian Telemental health service for early assessment among vulnerable occupational workers: mixed-method study with a user-centered design approach

PLOS One

Journal Requirements:

Reply: We confirm that our manuscript complies with all PLOS ONE style requirements.

Reply: We have added a subsection entitled “Ethical Considerations” to the manuscript, within the Methods section:

“In general, the study provided online informed consent to all participants, who took part voluntarily. Informed consent was signed virtually by participants by clicking on the “I agree to participate” option. Additionally, the identity of participants in the interviews, workshops, and surveys was protected through data anonymization and the custody of information with controlled access levels. Participants did not receive any type of monetary contribution for their participation.

This study was approved by the ethics and research committee of the National Institute of Health of Peru, through Directorial Resolution No. 206-2023-OGITT-INS (COD. OC-048-22).”

Reply: We confirm that this information has been updated in the submission and that both sections contain the same information.

We express our gratitude to the Spanish Agency for International Development Cooperation (AECID, in spanish) and the National Program for Scientific Research and Advanced Studies (PROCIENCIA-CONCYTEC-Peru, in spanish), who funded the project, enabling its execution, under contract number 076-2021-PROCIENCIA

Reply: We have made the requested modifications:

“Acknowledgments

Not applicable.

Funding Information

We express our gratitude to the Spanish Agency for International Development Cooperation (AECID, in spanish) and the National Program for Scientific Research and Advanced Studies (PROCIENCIA-CONCYTEC-Peru, in spanish), who funded the project, enabling its execution, under contract number 076-2021-PROCIENCIA.”

5. Please amend your list of authors on the manuscript to ensure that each author is linked to an affiliation. Authors’ affiliations should reflect the institution where the work was done (if authors moved subsequently, you can also list the new affiliation stating “current affiliation:….” as necessary).

Reply: We confirm that the names, affiliations, email addresses, and ORCID identifiers are correct in all cases.

6. Please remove your figures from within your manuscript file, leaving only the individual TIFF/EPS image files, uploaded separately. These will be automatically included in the reviewers’ PDF.

Reply: The images were included separately in TIFF format.

7. We note that Figure 1, Supplementary A, Supplementary B, Supplementary C, Supplementary D, in your submission contain copyrighted images. All PLOS content is published under the Creative Commons Attribution License (CC BY 4.0), which means that the manuscript, images, and Supporting Information files will be freely available online, and any third party is permitted to access, download, copy, distribute, and use these materials in any way, even commercially, with proper attribution. For more information, see our copyright guidelines: http://journals.plos.org/plosone/s/licenses-and-copyright.

a. You may seek permission from the original copyright holder of Figure 1, Supplementary A, Supplementary B, Supplementary C, Supplementary D, to publish the content specifically under the CC BY 4.0 license.

Reply: The figures and supplementary materials under review correspond to images that are the property of the authors of the present article; therefore, it is not necessary to request licenses from third parties. Likewise, it has been indicated in the “Supporting Information” section that the supplementary files are of original authorship.

Reply: We have included the captions for the Supporting Information files at the end of our manuscript.

“Supporting Information

Supplementary A. Early prototypes in co-design phase.

Supplementary B. TelePsico web platform prototype.

Supplementary C. Public information of the service.

Supplementary D. Telecare reports.

Supplementary material 1. Distribution of workers identified with mental health risk symptoms by job position

Supplementary material 2. Distribution of satisfaction and usability scores in external users (ESCOMA and SUS) according to sociodemographic characteristics.

Supplementary material 3. Distribution of usability rating in external users (SUS) according to sociodemographic characteristics.

Supplementary material 4. User Perceptions during Usability Evaluation through Interviews.

Supplementary material 5. Service satisfaction qualification levels by ESCOMA instrument D1 and D3 dimensions

Supplementary material 6. Perceptions of external and internal users regarding the acceptability of the service.”

Reply: Ok.

Reply: We have reviewed the reference list and confirm that it is correct.

Review Comments to the Author

Reviewer #1: Thank you for the opportunity to review this paper. The topic of a national-scale telehealth application for mental health is highly significant. The study is exceptionally well-designed and executed, demonstrating a broad scope and resulting in a practical and currently implemented system. It is commendable that the system is already operational and in use, and I have reviewed the corresponding website.

I have a few specific comments aimed at further strengthening the manuscript:

Reply: Thank you for your comments. We will respond to each one.

1- I recommend a thorough check of the manuscript's punctuation and grammar. For instance, in the Abstract (Background section), I noted a sentence concluding with two periods (..) instead of a single one. A general review for consistency in punctuation (commas, periods, colons, etc.) across the entire text is advised.

Reply: Thank you for your recommendation. We will respond to each one.

2- In the Introduction section, while the third paragraph makes an essential reference to prevalence data, the Introduction would be significantly improved by explicitly stating the global prevalence of mental health disorders, followed by specific prevalence statistics within Peru. The current text implies the source of the data through citations, but it should explicitly state in the text that these statistics pertain to Peru.

Reply: As global prevalence data are already presented in the first paragraph of the Introduction, we have revised the third paragraph to explicitly state that the reported prevalence figures correspond to the Peruvian context.

3- In the Method section, the description of the setting for the system's implementation is currently ambiguous. If the scope is indeed national (national scale), this should be stated explicitly and specifically in the Methods section to reflect the significant scale of the project.

Reply: We have revised the Methods section to explicitly clarify that the digital mental health system was implemented at a metropolitan scale, specifically in Lima and Callao.

4- In the Result section, the inclusion of a flowchart (Figure 2) and a view of the initial website page in the Appendix is helpful. However, to better illustrate the user experience and the functional processes outlined in Figure 2 (from screening to the final report), it is recommended to include additional screenshots of the live system/website in the main body of the paper.

Reply: Thank you for your recommendation. We will respond to each one.

5- The Discussion is well-written, thoroughly analyzing the work from various angles. To provide richer context and academic rigor, it is suggested to integrate relevant similar studies in the appropriate sections and include a comparative analysis of this work against those studies.

Reply: We have revised the Discussion section to more explicitly integrate relevant comparable studies and to include a clearer comparative analysis between our findings and previously published telehealth interventions.

6- If this information is not planned for a separate publication, it would be beneficial to include some current key utilization metrics for the operational system. This could consist of the total number of individuals who have received assistance from the system to date, or other relevant indicators of current scale and impact.

Reply: At the time of manuscript submission, key utilization indicators of the TelePsico CENSOPAS system—specifically the number of workers evaluated and sensitized—were available and have already been reported in the Results section under Telehealth service intervention outcomes.

Reviewer #2: This manuscript addresses a timely and significant issue regarding mental health support for vulnerable occupational groups during the COVID-19 pandemic. The application of a User-Centered Design (UCD) framework is a major strength of this study, providing a robust model for developing effective health IT solutions. However, there are specific methodological clarifications and minor corrections required before publication.

Reply: Thank you for your comments. We will respond to each one.

1) Methodological Terminology:In the "Usability, satisfaction and acceptability assessment" section, the analysis is described as a "phenomenological approach." Given that the results focus on usability feedback and satisfaction rather than the deep essence of lived experiences, this term seems misapplied. The analysis appears to align more closely with "Thematic Analysis" or "Qualitative Content Analysis." Please revise this terminology to accurately reflect the method used.

Reply: We thank the reviewer for this valuable comment. After extensive discussion among the authors, we agree that the reviewer’s comment is correct. Our study focuses on usability- and satisfaction-related comments; therefore, it is better aligned with a thematic analysis. We have revised the text throughout the manuscript to specify that the analysis is thematic.

2) Citation Accuracy: Reference [45] (Universitat Oberta de Catalunya, Interview Toolkit) is currently cited in the context of data analysis using Atlas Ti. This reference pertains to data collection (interviews), not analysis. Please correct this citation to a source relevant to the analytical method employed.

Reply: Thank you for your comments. We will respond to each one.

3) Abstract Clarity: The Abstract mentions "81 participants in the efficacy assessment" without sufficient context. To avoid confusion regarding the response rate, please clarify that these 81 participants are a subset of the 134 users who received psychoeducation.

Reply: The Abstract was revised in accordance with the reviewer’s suggestion to improve clarity.

4) Comparative Analysis Context: The study compares satisfaction scores with historical outpatient data (Moscoso et al.). While valid as a benchmark, please explicitly acknowledge in the Discussion or Limitations section that comparing a telehealth service (during a pandemic) with traditional outpatient services (pre-pandemic) involves different contexts and user expectations.

Reply: We have made the requested modifications:

“The level of service usage satisfaction, evaluated using the ESCOMA instrument, showed a high score in satisfaction with care processes and psychological care. The evaluation instrument was initially developed by Moscoso et al., who found lower scores in satisfaction levels among users of MINSA, EsSalud, the Armed Forces, and private entities who received outpatient medical care services. These differenc

---

## [Decision Letter · Decision Letter 1]

8 Feb 2026

Evaluation of usability and acceptability of a Peruvian Telemental health service for early assessment among vulnerable occupational workers: mixed-method study with a user-centered design approach

PONE-D-25-40617R1

Dear Dr. Villarreal-Zegarra,

We’re pleased to inform you that your manuscript has been judged scientifically suitable for publication and will be formally accepted for publication once it meets all outstanding technical requirements.

Kind regards,

Jahanpour Alipour, Ph.D.

Academic Editor

PLOS One

Additional Editor Comments (optional):

Reviewers' comments:

Reviewer's Responses to Questions

**Comments to the Author**

Reviewer #1: All comments have been addressed

Reviewer #2: (No Response)

2. Is the manuscript technically sound, and do the data support the conclusions?

Reviewer #1: Yes

Reviewer #2: (No Response)

3. Has the statistical analysis been performed appropriately and rigorously?

Reviewer #1: Yes

Reviewer #2: (No Response)

4. Have the authors made all data underlying the findings in their manuscript fully available?

Reviewer #1: Yes

Reviewer #2: (No Response)

5. Is the manuscript presented in an intelligible fashion and written in standard English?

Reviewer #1: Yes

Reviewer #2: (No Response)

Reviewer #1: I would like to thank the authors for their thorough responses and the effort put into revising the manuscript. I am pleased to note that all of my previous concerns have been addressed satisfactorily, and I have no further comments. I believe the paper is now ready for publication.

Reviewer #2: The authors have satisfactorily addressed the previous comments and revised the manuscript to meet the journal's publication criteria. The methodological terminology has been corrected, the scope of the study is accurately defined, and the limitations regarding the comparative analysis are now appropriately contextualized. I have no further concerns.

**Do you want your identity to be public for this peer review?** For information about this choice, including consent withdrawal, please see our Privacy Policy

Reviewer #1: No

Reviewer #2: No

---

## [Editor Report · Acceptance letter]

PONE-D-25-40617R1

PLOS One

Dear Dr. Villarreal-Zegarra,

I'm pleased to inform you that your manuscript has been deemed suitable for publication in PLOS One. Congratulations! Your manuscript is now being handed over to our production team.

Kind regards,

on behalf of

Dr., Jahanpour Alipour

Academic Editor

PLOS One